# Changes in Phenolics during Cooking Extrusion: A Review

**DOI:** 10.3390/foods10092100

**Published:** 2021-09-05

**Authors:** Evžen Šárka, Marcela Sluková, Svatopluk Henke

**Affiliations:** Department of Carbohydrates and Cereals, University of Chemistry and Technology, Prague, Technicka 5, 166 28 Prague, Czech Republic; marcela.slukova@vscht.cz (M.S.); henkes@vscht.cz (S.H.)

**Keywords:** extrusion cooking, total phenolics, flavonoids, phenolic acids, total phenolics, retention

## Abstract

In this paper, significant attention is paid to the retention of phenolics in extrudates and their health effects. Due to the large number of recent articles devoted to total phenolic content (TPC) of input mixtures and extrudates, the technological changes are only presented for basic raw materials and the originating extrudates, and only the composites identified has having the highest amounts of TPC are referred to. The paper is also devoted to the changes in individual phenolics during extrusion (phenolic acids, flavonoids, flavonols, proanthocyanidins, flavanones, flavones, isoflavons, and 3-deoxyanthocyanidins). These changes are related to the choice or raw materials, the configuration of the extruder, and the setting the technological parameters. The results found in this study, presented in the form of tables, also indicate whether a single-screw or twin-screw extruder was used for the experiments. To design an extrusion process, other physico-chemical changes in the input material must also be taken into account, such as gelatinization of starch; denaturation of protein and formation of starch, lipids, and protein complexes; formation of soluble dietary fiber; destruction of antinutritional factors and contaminating microorganisms; and lipid oxidation reduction. The chemical changes also include starch depolymerization, the Maillard reaction, and decomposition of vitamins.

## 1. Introduction

As a thermomechanical process, extrusion cooking is used for the production of a large number of food products, e.g., breakfast cereals [1], confectionery products, biscuits [2], ready-to-eat expanded snacks, meat analogues [3,4], modified starches, and pet foods [2]. The extrusion-cooking technique is also suitable for the production of specific precooked or gluten-free pasta [5,6]. Meat analogues are commonly produced from protein sources such as soy, wheat, pea, milk, egg, and fungal substrates by applying high-moisture extrusion technology [7]. Extrusion can also be used as a pre-processing intervention to improve the functionality of a material as a food ingredient. Many consumers now accept the slightly different taste and texture of extruded products, particularly if the food is healthier.

The extrusion process results in numerous transformations, such as gelatinization of starch, denaturation of proteins [2], formation of slowly digestible and resistant starch [8], disintegration of antinutritional factors, and increase in soluble dietary fiber [9].

The main ingredient of extrudates is usually cereal flour (corn grits, rice flour, wheat flour, barley grits, or sorghum flour). Oat, chickpea, soybean, white ginseng root hair, cassava flour, germinated *Chenopodium*, dried mango, tigernut flour, dried green banana flour, and millet have also been tested (see the following text and tables), often with additives, which are usually materials rich in phenolics and/or waste from the food industry.

The two basic types of extruders are single- and twin-screw mechanisms, in which the screws co- or counter rotate. The temperature and pressure in the extruder are not only influenced by the shape of the screw, the technological parameters (mass and heat flow), and the processed material, but the die design (particularly the die diameter) is also highly important. To ensure high porosity of the extrudates, a high temperature is required, usually from 140 to 160 °C [8]. Operation limits also exist for extrusion processing, such as water content of the mixture, pH, fat content, stickiness of the mixture, high temperatures in zones of the extruder (causing color changes and significant changes in the content of vitamins) and particle size. In addition, economic, hygienic, and technological benefits must also be taken into account.

Polyphenols are secondary metabolites of plants that are used in their defense against severe environments, such as ultraviolet radiation or attack by pathogens [10]. These compounds are generally classified as flavonoids, phenolic acids, lignans, and stilbenes. They are widely distributed in plant-based food products and influence the Maillard reaction [11]. Phenolics are linked to several health benefits, including antioxidant, antibacterial, antiglycemic, antiviral, anticarcinogenic, anti-inflammatory, and vasodilatory properties [12].

Phenolic compounds have two forms—free and bound. Bound phenolic compounds are often bound with cell wall components [13]. Decomposition of heat-labile phenolic compounds and polymerization of some phenolic compounds during extrusion at high temperatures tend to decrease the extractable phenolic content. However, lowering the temperature can cause a decrease in the porosity and taste perception of the extrudates. Conversely, however, due to the disruption of cell wall matrices and the breaking of high molecular weight complex phenolics during extrusion at high temperatures, the extractability and solubility of phenolic compounds is improved [14]. The extractability can be supported by previous germination [15]. Extrusion cooking induces an alteration in physicochemical and functional properties, and the phenolic compounds content and their antioxidant activity, which also depends on the raw material and numerous variables, such as feed moisture, screw speed, and configuration of the extruder, i.e., die and screw geometry, temperature, and time [16].

Hirth et al. [17] showed that the destruction of phenolics during extrusion typically occurs in first-order reactions. A description of a kinetic evaluation was also published in the paper of Xu et al. [18]. Additionally, several authors have reported that phenolic compounds influence carbohydrate hydrolysis [19,20]. A variety of phenolic compounds have been shown to inhibit the activities of α-amylase and/or α-glucosidase. The inhibitory phenolic compounds include flavonoids (flavanones, anthocyanins, flavanols, isoflavones, and flavones), phenolic acids, and tannins (proanthocyanidins and ellagitannins). Phenolic compounds are postulated to bind to active or secondary sites of digestive enzymes [21] and/or bind to substrate, thus reducing starch hydrolysis. The lower digestibility of starch results in lower obesity and microbial formation of short chain fatty acids in the human gut, thus providing health effects [22].

The total phenolic content of extrudates is stated in numerous scientific papers. Thus, in the current study we focused more on individual groups of phenolics (phenolic acids, flavonoids, flavonols, proanthocyanidins, flavanones, flavones, isoflavons, 3-deoxyanthocyanidins, etc.) and their changes during extrusion cooking. The object of the study was to compare analytical results in premixes and extrudates (retention of phenolics). We are aware that mixing results of TPC/TFC with other specific determinations (chromatography/spectrometry coupled methods) is not always correct, but the data provide basic information for new food design. The paper does not address the suitability of the analytical methods used.

Application of one-screw or twin-screw extrusion is also investigated in this review. The detailed arrangement of the instruments and sensory properties of the products can be found in the reference papers.

## 2. Total Phenolic Content (TPC) in Raw Materials and Extrudates

The first step in the design of a new product should be trials of the extrusion processing for a one-component blend, although the input blend clearly does not have a single component (usually cereal flour); the second component is water. Sensory properties of the resulting extrudates (crispness, hardness) depend foremost on the quantity of water used. The single component data enables the impacts on TPC to be assessed in real mixtures.

The TPC data of one-component extrudates from recent years are included in Table 1 (incl. references). Some of these were selected from papers focused on multi-component blends in which a one-component extrudate was used as a standard. The data given in Table 1 can therefore be useful for researchers designing a new composition of the input mixture for cooking extrusion.

Corn extrudates have special properties and are of economic importance in the food industry. It is clear from Table 1 that the TPC found in these extrudates is very low, in the range of 34–132 mg/100 g DM (dry matter). Lime-cooking extrusion represents an alternative technology for manufacturing pre-gelatinized flours for tortillas, with the advantages of saving energy and no generation of effluents. The phytochemical profiles were studied by Aguayo-Rojas et al. [23] and Gaxiola et al. [24]. The TPC in tortillas after lime-cooking extrusion was found to be higher, ranging from 111 to 204 mg/100 g DM.

Many data have been published for rice flour, particularly for brown rice. Different results of TPC for extrudates prepared by a single-screw extruder compared to a twin-screw extruder were found in the ranges of 94–178 and 46–77 mg/100 g DM, respectively. It is likely that stronger conditions in a single-screw extruder caused higher disruption of cell wall matrices. Besides extrusion conditions, processing temperature and screw speed are also important. Other data were measured for flour from polished rice, broken rice, glutinous rice, glutinous rice with α-amylase addition, extruded soaked rice, and extruded germinated rice. As expected, the highest value was found for rice bran—811 mg/100 g DM.

The data for wheat in Table 1 may appear to show the opposite pattern, i.e., a higher value was found in flour than in bran. This may be due to the analytical data of some of the cited article(s), because bran usually has greater TPC. Nonetheless, TPC retention for wheat bran was found to be higher than 100%, similar to that of rice bran. The retention of wheat TPC for ground wheat is similar to the retention of barley grits as published by Sharma et al. [25].

Sorghum is a gluten-free cereal that has the highest content of phenolic compounds among cereals but must be processed prior to use for human consumption. The phenolic compounds of sorghum, including proanthocyanidins, 3-deoxyanthocyanidins, and flavones, beneficially modulate the gut microbiota and variables related to noncommunicable diseases, such as obesity, diabetes, dyslipidemia, cardiovascular disease, and cancer [26]. TPC in sorghum depends foremost on its variety. TPC after extrusion was found to be within the range 180–1765 mg/100 g DM.

Other data in Table 1 relate to the TPC of oat, chickpea, soybean, white ginseng root hair, cassava flour, germinated *Chenopodium*, dried mango, tigernut flour, dried green banana flour, and millet. The highest TPC values have been found in extrudates of oven-dried green banana flour, tigernut flour, and soybean flour, having values of 1196, 1000, and 652 mg/100 g DM, respectively.

Numerous TPC data of extrudates are presented in the literature for which, in addition to water and one main component, additives were also used, e.g., materials rich in phenolics and/or waste from the food industry. This procedure enables important phenolics in extrudates to be increased. Therefore, we also complied interesting TPC data for blends comprising a greater number of components and their extrudates in Table 1. More data for multi-component blends can be found in our previous paper [26]. TPC values of 3285–4038 and 1522–3160 mg/100 g DM were found in extrudates of the mixtures of soy protein isolate/wheat gluten/corn starch/green tea and lentil flour/orange peel powder, respectively. The highest TPC retention was found in rice flour/goji berries and genetically-modified corn/bean grits, having values of up to 184% and 174%, respectively. Conversely, low retention was found in soy protein isolate/wheat gluten/corn starch, broken rice/açaí, and corn grits/carrot, in the range from 36 to 57%. The other mentioned alternative to increase TPC is the germination of seeds, enzyme processing, or the optimization of the extrusion process.

## 3. Phenolic Acids

Phenolic acids are a highly important component of phenolics, e.g., phenolics present in cereals are mostly phenolic acids covalently bound to the cell wall, together with carotenoids, tocopherols, and tocotrienols [27]. Newly published data relating to the total content of phenolic acids are surveyed in Table 2. The data show extrusion cooking can cause the degradation of heat labile phenolic acids or promote their interaction with the nutrients released from the food matrix. This can result in a decrease in phenolic acids. In contrast, extrusion cooking can also disrupt the cell wall matrices and cleave the covalent bonds between the cell wall polysaccharides and bound phenolic acids, leading to the release of phenolic acids, which are extractable [28]. Practically no changes in the total quantity of phenolic acids (Table 2) were found after extrusion, with the exception of oat flour having retention of 145% [29].

New laboratory techniques such as (ultra) high-performance liquid chromatography (HPLC) electrospray ionization tandem (quadrupole time-of-flight) mass spectrometry, and the simple HPLC system, have enabled phenolic acids to be separately identified.

The individual data from the literature are also numerous. Therefore Table 3 presents only a selection of these findings. A short comment on some of the papers is provided the following text. Wojtowicz et al. [30] tested the application of Moldavian dragonhead (*Dracocephalum moldavica* L.) leaves in extruded corn snacks. The data of the individual phenolic acids from the literature in Table 2 show that the maximum content was only found for rosmarinic acid, which showed a high antioxidant potential and a radical scavenging activity in extruded corn snacks containing Moldavian dragonhead leaves, particularly when a high content of additive was used.

The largest increase (retention) was found for sinapic acid in various materials: 262% in oat flour, 275% in corn grit, and 405% in brown rice flour. However, the final concentration in extrudates was not high: between 2 and 9 mg/100 g DM. Although some authors have indicated that wine is a good source of sinapic acid, the concentration is only about 0.2–0.4 mg/L [31], and higher content has been found in rapeseed seeds—84 mg/100 g DM [32]. Sinapic acid has been reported to provide benefits against various pathological conditions, such as infections, oxidative stress, inflammation, cancer, diabetes, neurodegeneration, and anxiety [33].

Zhang et al. [34] detected seven phenolic acids in rice fractions, namely, ferulic, vanillic, p-coumaric, chlorogenic, gallic, caffeic, and syringic acids. The largest values of phenolic acids in extrudates were found in rice bran, and the highest content in gallic acid. According to Sun et al. [35], gallic acid is a strong antioxidant and antimutagenic and anticarcinogenic agent, and it is attractive due to its high content in blend extrudates. The content of vanillic acid found in rice bran extrudates—45 mg/100 g DM (having retention of 199%)—is significantly higher than that of, for example, honey, which is 0.07–0.19 mg/100 g [36]. Kim et al. [37] suggested that vanillic acid may be a useful therapeutic candidate for ulcerative colitis.

Furthermore, Zeng et al. [29] and Gong et al. [38] also investigated phenolic acids in brown rice flour, but found significantly different results for vanillic and caffeic acids. Zeng et al. [29] found retention of 266% of caffeic acid in extrudates of brown rice flour, resulting in very low content, of 0.26 mg/100 g DM. For comparison, thyme, sunflower seeds, and lingonberry have 20, 8, and 6 mg/100 g, respectively [39]. Caffeic acid is an antioxidant, and shows immunomodulatory and anti-inflammatory activity. Zeng et al. [29] also found high retention, of 196%, for syringic acid in wheat flour extrudates having 0.5 mg/100 g DM. Syringic acid is often found in fruits and vegetables, and the value for extrudates is near the value in strawberries, of 0.5 mg/100 g [40]. Syringic acid shows a wide range of therapeutic applications in the prevention of diabetes, CVDs, cancer, cerebral ischemia. In addition, it possesses anti-oxidant, antimicrobial, anti-inflammatory, antiendotoxic, and neuro- and hepatoprotective activities. It has an effective free radical scavenger and alleviates the oxidative stress markers [41].

4-OH-benzoic acid shows antimicrobial activity, and has significant retention (214%) in oat flour extrudates where the final content was about 1 mg/100 g DM. This value is very low compared with the uncharacteristic predominance of 4-OH-benzoic acid in the mesocarp of *Cocos nucifera*, having 44 mg/100 g DM [42].

Lohani and Muthukumarappan [43] extruded a mixture containing corn flour, sorghum flour, and apple pomace, and revealed that major phenolic acids in extruded products were derived by caffeic acid, followed by salicylic acid and ferulic acid. Thakur et al. [44] studied the extrusion behavior of grits obtained from three successive reductions produced by dry milling two normal corn types and one waxy corn at different extrusion temperatures. The grit from each reduction stage showed the presence of base-hydrolyzed bound protocatechuic acid, p-coumaric acid, sinapic acid, and ferulic acid. The concentrations of these acids decreased after acid hydrolysis, whereas that of gallic acid increased. Diferulic acid and 4-OH-benzoic acid were identified by Gong et al. [38] in brown rice flour.

Kasprzak et al. [66] prepared blends of corn grits and 2%, 4%, 6%, and 8% of *Brassica oleracea* L. (kale). Both the qualitative and quantitative contents of phenolics increased with the addition of kale; 3-OH-cinnamic acid occurred only in snacks enriched with 6% kale. Oniszczuk et al. [67] processed by grinding dehulled and roasted buckwheat grains, and detected the following in the obtained samples: gallic, protocatechuic, gentisic (5-OH-salicylic), 4-hydroxybenzoic, vanillic, trans-caffeic, cis-caffeic, trans-p-coumaric, cis-p-coumaric, syringic, trans-ferulic, cis-ferulic, salicylic, trans-sinapic, and cis-sinapic acids. Guven et al. [52] investigated the effect of the extrusion process on the bioactive compounds and in vitro bio-accessibility of cynarin and cynaroside, which are found in artichoke leaf powder. Cynarin (1,3-di-O-caffeoylquinic acid) is known to prevent cholesterol biosynthesis and low-density lipoprotein oxidation.

## 4. Flavonoids, Flavonols, Proanthocyanidins, Flavanones, Flavones, Isoflavons, and 3-Deoxyanthocyanidins

Significant degrees of antioxidant and anti-tumor activity and the converse, carcinogenicity, have been attributed to flavonoids, and it is their effect on human health which has brought them to wider attention. The basic structure of the major groups of flavonoids can be found in the literature [70]. Table 4 presents the total flavonoid content (TFC) from the referred papers. The highest and lowest values of total flavonoid content surveyed in Table 4 were found in extrudates prepared from rice bran and corn, respectively. TFC in unprocessed rice bran was much higher than that in polished and brown rice, which indicates that phenolics are more concentrated in the bran fraction of rice. After extrusion, TFC fell by 40 and 30% in polished rice and in brown rice, respectively [34]. Rathod and Annapure [62] revealed a very high nutritional value of the extruded product of lentil flour and orange peels with the retention of total flavonoid content of 67%.

The most well-known flavonoids are anthocyans, flavanons, flavonols, and chalcones [71]. Flavonols, which are flavonoids with the 3-hydroxyflavone backbone, are the most ubiquitous flavonoids in foods. They are present in onions at concentrations up to 120 mg/100 g, but also in kale, leeks, broccoli, and blueberries [72]. Newly studied data of total flavonols in the extrusion process are shown in Table 5.

The content of total flavonols after extrusion increased in apple pomace [68] but, conversely, in lentil flour decreased with retention of only 13% [73]. The highest content was determined in extrudates prepared from corn and red potatoes—33 mg/100 g DM [45]. The highest natural content of flavonols in vegetables was found in red onions (45.25 mg/100 g) and dill (42.09 mg/100 g) [74]. However, Pastor-Villaescusa et al. [75] warned that foods containing sufficient flavonols tend to be high in calories; thus, consumption must be carefully controlled.

The phenolic profile of flavonols was determined using the Dionex Ultimate 3000 UPLC (Thermo Scientific, San Jose, CA, USA), where data were collected simultaneously with a diode array detector (280 and 370 nm) and a mass spectrometer (Linear Ion Trap LTQ XL mass spectrometer, Thermo Finnigan, San Jose, CA, USA) operating in negative mode [73,76]. The data of individual flavonols are surveyed in Table 6.

According to Liu et al. [68], quercetin, an important flavonol, increased after extrusion. Ciudad-Mulero et al. [76] found the highest content of catechin hexoside in extrudates of lentil flour and yeast, of up to 6 mg/100 g DM. For comparison, Gonçalves et al. [77] noted that, in addition to quercetin-3-O-rutinoside, 0.39–26.55 mg/100 g of quercetin-3-O-glucoside was found in sweet cherries. Previous studies noted that quercetin commonly found in onions and apples may increase bone formation; suppress bone resorption by decreasing the differentiation of osteoclast progenitor cells and inhibiting the activity of mature osteoclasts; and increase bone minerals in rats [78].

Proanthocyanidins are oligomers or polymers of flavan-3-ols and are often found in fruits, berries, beans, nuts, cocoa, and wine. Proanthocyanidins are considered to have a wide range of biological properties, including antioxidant, anticarcinogenic, cardioprotective, antimicrobial, and neuroprotective activities, as demonstrated by a number of in vitro and in vivo studies [79]. Cardoso et al. [80] identified proanthocyanidins in sorghum extrudates, although only in genotype SC391, in which the content was reduced by extrusion cooking by 52%. They also found that, in the extruded genotypes of sorghum, flavanones (naringenin and eriodictyoland) and flavones (luteolin and apigenin) were totally destroyed after extrusion cooking (100%) due to their thermal sensitivity. The 3-deoxyanthocyanidins (luteolinidin, apigeninidin, 5-methoxy-luteolinidin, and 7-methoxy-apigeninidin) were also highly susceptible to extrusion cooking.

Azad et al. [56] prepared soybean food composite by hot melt extrusion (STS-25HS twin-screw extruder; Hankook E.M. Ltd., Pyoung Taek, Korea) from hydrophilic food-grade hydroxypropyl methylcellulose and soybean, and determined the isoflavone content (daidzin, daidzein, glycitein, glycitin, genistein, and genistin). It has been reported that genistein, the major nutraceutical found in soybean and its products, has health benefits, including prevention of cancer, cardiovascular diseases, obesity, and osteoporosis, and attenuation of postmenopausal problems [81].

## 5. Conclusions

Extrusion cooking induces the alteration of raw materials in physicochemical and functional properties, phenolic compounds, and their antioxidant activities. Decomposition of heat-labile phenolic compounds and polymerization during extrusion tend to decrease the extractable phenolic content. In contrast, due to the disruption of cell wall matrices and the breaking of high molecular weight complex phenolics during extrusion, the extractability of phenolic compounds is improved.

Numerous interesting data relating to phenolics have been published for rice flour, particularly that of brown rice, and composite mixtures. The largest values of phenolic acids in extrudates were found in rice bran. Composite mixtures include, e.g., addition of orange peel powder, green tea, goji berries, bean grits, açaí, and carrot. High levels of TPC, of 3285–4038 and 1522–3160 mg/100 g DM, were found in extrudates of soy protein isolate/wheat gluten/corn starch/green tea and lentil flour/orange peels powder, respectively.

Regarding phenolic acids, excellent sources of vanillic acid and syringic acid are extrudates based on rice bran or wheat flour. The use of additives in the cereal premix, e.g., *Dracocephalum moldavica*, *Cocos nucifera*, *Brassica oleracea*, red potatoes (*Solanum tuberosum* var. Magenta Love), strawberries (*Fragaria*
*ananassa*), and apple pomace, can achieve even higher concentrations of selected phenolic acids.

Extrusion reduces flavonoids by 40% and 30% in polished rice and brown rice, respectively. The flavonol content after extrusion was found to increase in apple pomace, but decrease in lentil flour with retention of only 13%. Regarding flavonol content in extrudates, we recommend that new experiments should be conducted with rice or lentil flour mixed with parsley or dill. Flavanones, flavones, and 3-deoxyanthocyanidins in the extruded genotypes of sorghum are destroyed after extrusion cooking due to their thermal sensitivity.

Thus, extrusion cooking can improve or worsen the digestibility of phenolic compounds. Higher values of specific phenolic compounds in extrudates create the potential for new food products.

## Figures and Tables

**Table 1 foods-10-02100-t001:** Total phenolic content (TPC) in the raw material and extrudates from different food materials.

Source	TPC Content in Input Raw Material (mg/100 g DM)	TPC Content in Output Extrudates (mg/100 g DM)	Retention (%)	Type of Extruder	Reference
corn grit	-	132 ^1^, 34 ^2^	-	single-screw	[45] ^1^, [46] ^2^
corn grit	-	41	-	co-rotating twin-screw	[47]
corn grit	126–165 ^1^, 233–244 ^2^	111–128 ^1+^, 186–204 ^2+^	74–87 ^1^, 80–84 ^2^	single-screw; nixtamalization	[23] ^1^, [24] ^2^
brown rice flour	132 ^1^, 253 ^2^, 93 ^3^	129 ^1^, 178 ^2^, 94 ^3^	98 ^1^, 70 ^2^, 101 ^3^	single-screw	[38] ^1^, [48] ^2^, [49] ^3^
brown rice flour	178 ^1^	77 ^1^, 47–50 ^2^, 46 ^3^	43 ^1^	twin-screw	[38] ^1^, [50] ^2^, [29] ^3^
flour from polished rice	57	26	46	co-rotating twin-screw	[38]
flour from broken rice	44	30	68	single-screw	[51]
flour from glutinous rice	36	7–21	20–57	co-rotating twin-screw	[18]
flour from glutinous rice + α-amylase	36	12–31	34–85	co-rotating twin-screw	[18]
extruded soaked rice	98	99	101	single-screw	[49]
extruded germinated rice	141	135	96	single-screw	[49]
rice bran	756	811	107	co-rotating twin-screw	[38]
wheat flour	340	220	65	co-rotating twin-screw	[52]
ground wheat	266	210	79	single-screw	[29]
wheat bran, var. Kronstad	238	248–384	104–161	single-screw	[53]
wholemeal sorghum flour, vars. Macia, NK 283, Gadam El Haman, PAN 8629r	270–2450 ^1^; 2335–2834 ^2^	180–670 ^1^; 580–1765 ^2^	30–67 ^1^; 24–62 ^2^	co-rotating twin-screw	[54] ^1^, [55] ^2^
decorticated sorghum flour, vars. Macia and NK 283	200–850	230–410	38–170	co-rotating twin-screw	[54]
oat flour	158	146	92	single-screw	[29]
chickpea	-	60	-	co-rotating twin-screw	[47]
soybean	540	652–660	120–122	twin-screw	[56]
white ginseng root hair	278	292–553	105–199	twin-screw	[57]
cassava flour	570	370	64	single-screw	[58]
germinated *Chenopodium* flour	565	57–81	10–14	twin-screw	[59]
dried mango	-	267–510	-	single-screw	[60]
tigernut flour	1130	1000	88	single-screw	[58]
millet	295	455	154	twin-screw	[56]
oven dried green banana flour	338	1196	356	co-rotating twin-screw	[61]
lentil flour and orange peels powder	4685	3285–4038	70–86	co-rotating twin-screw	[62]
soy protein isolate/wheat gluten/corn starch/green tea	3847–5749	1522–3160	39–55	co-rotating twin-screw	[63]
rice flour/goji berries	123–402	51–741	41–184	twin-screw	[4]
genetically modified corn (line 1041/1.7k) and bean grits (var. Jamapa Black)	140	244	174	single-screw	[64]
soy protein isolate/wheat gluten/corn starch	2747	985	36	co-rotating twin-screw	[63]
broken rice/açaí	188–500	104–279	51–56	single-screw	[51]
corn grits and carrot	1820	950–1030	52–57	co-rotating twin-screw	[65]

^+^ in tortillas; the exponents refer to the literature in the last column; if the variety is not listed in the table, then either the data refer to more than one variety or the variety was not listed in the article; “-” means no measured data.

**Table 2 foods-10-02100-t002:** Total content of phenolic acids in the raw material and extrudates from different food materials.

Source	Total Content of Phenolic Acids in Input Raw Material (mg/100 g DM)	Total Content of Phenolic Acids in Output Extrudates (mg/100 g DM)	Retention (%)	Type of Extruder	Reference
corn grit	-	24	-	single-screw	[45]
corn grit	202–215	200–213 ^+^	99	single-screw; nixtamalization	[24]
brown rice flour	36 ^1^, 40 ^2^	43 ^1^, 42 ^2^	121 ^1^, 106 ^2^	single-screw	[38] ^1^, [29] ^2^
brown rice flour	77	71	92	twin-screw	[34]
rice bran	217	222	102	co-rotating twin-screw	[34]
flour from polished rice	26	23	89	co-rotating twin-screw	[34]
ground wheat	60	63	104	single-screw	[29]
oat flour	27	40	145	single-screw	[29]

^+^ in tortillas; the exponents refer to the literature in the last column; DM—dry matter; if the variety is not listed in the table, then either the data refer to more than one variety or the variety was not listed in the article; “-” means no measured data.

**Table 3 foods-10-02100-t003:** Phenolic acids in the raw material and extrudates from different food materials.

Phenolic Acid	Source	Content in Input Raw Material (mg /100 g DM)	Content in Output Extrudates (mg /100 g DM)	Retention (%)	Reference
Gallic acid	rice bran	7	6	91	[34]
	brown rice flour	5	3	74	[34]
	buckwheat	0.3	0.2–0.3	60–98	[67]
	apple pomace, cv. Pink Lady	0.17	Nd	-	[68]
Vanillic acid	corn grit	2–3	3	103–142	[24]
	rice bran	23	45	199	[34]
	brown rice flour	23 ^1^, 2 ^2^, 0.3 ^3^	26 ^1^, 32 ^2^, 0.4 ^3^	111 ^1^, 125 ^2^, 118 ^3^	[34] ^1^, [38] ^2^, [29] ^3^
	wheat flour	0.6	0.7	105	[29]
	oat flour	0.7	1.1	150	[29]
	buckwheat	0.07	0.03–0.07	47–99	[67]
	corn/kale, var. sabellica	-	0.08–0.11	-	[66]
Syringic acid	corn grit	0.9–1.6	1.5–2.2	137–167	[24]
	rice bran	3.3	4.3	130	[34]
	flour from polished rice	Nd	Nd	-	[34]
	brown rice flour	1.1 ^1^, 0.7 ^2^, 0.4 ^3^	1.0 ^1^, 0.7 ^2^, 0.4 ^3^	88 ^1^, 103 ^2^, 100 ^3^	[34] ^1^, [38] ^2^, [29] ^3^
	wheat flour	0.2	0.5	196	[29]
	oat flour	0.6	1.1	173	[29]
Ferulic acid	corn grit	168–186	167–179	96–99	[24]
	rice bran	166	152	91	[34]
	brown rice flour	39 ^1^, 44 ^2^, 31 ^3^	34 ^1^, 53 ^2^, 31 ^3^	86 ^1^, 119 ^2^, 100 ^3^	[34] ^1^, [38] ^2^, [29] ^3^
	wheat flour	46	50	109	[29]
	oat flour	19	28	143	[29]
	corn/kale, var. sabellica	-	0.2–4.6	-	[66]
	corn flour/sorghum flour/apple pomace	4–5	2.5–3.5	71–76	[43]
	rice/pea	-	1.2–3.4	-	[69]
	apple pomace, cv. Pink Lady	0.06	0.04–0.08	67–133	[68]
4-OH-Benzoic acid	brown rice flour	0.8 ^1^, 0.4 ^2^	1.0 ^1^, 0.6 ^2^	114 ^1^, 146 ^2^	[38] ^1^, [29] ^2^
	wheat flour	4.4	2.5	57	[29]
	oat flour	0.4	0.9	214	[29]
	buckwheat	0.02	0.01–0.03	62–137	[67]
	rice/pea	-	0.03–0.07	-	[69]
	corn/kale, var. sabellica	-	0.01- 0.03	-	[66]
	corn flour/sorghum flour/apple pomace	1.7–2.3	1.1–1.7	63–71	[43]
Rosmarinic acid	corn/dragonhead leaves	-	287–2884	-	[30]
5-OH-Salicylic acid	buckwheat	0.02	0.02–0.04	131–248	[67]
Sinapic acid	corn grit	2–4	7–9	209–275	[24]
	corn/kale, var. sabellica	-	0.6–1.3	-	[66]
	brown rice flour	4 ^1^, 0.5 ^2^	7 ^1^, 2.2 ^2^	197 ^1^, 405 ^2^	[38] ^1^, [29] ^2^
	wheat flour	1.0	1.8	186	[29]
	oat flour	0.7	1.9	262	[29]
Chlorogenic acid	rice bran	11	8	70	[34]
	brown rice flour	3.3 ^1^, 1.8 ^2^, 0.73 ^3^	2.6 ^1^, 1.9 ^2^, 1.0 ^3^	79 ^1^, 105 ^2^, 134 ^3^	[34] ^1^, [38] ^2^, [29] ^3^
	wheat flour	0.7	0.9	121	[29]
	oat flour	1.9	1.9	100	[29]
	apple pomace, cv. Pink Lady	3	5–6	149–163	[68]
	corn flour/sorghum flour/apple pomace	0.3–0.8	0.05–0.3	15–38	[43]
	corn/red potato, var. Magenta Love	-	3–45	-	[46]
	corn/purple potato, var. Blue Star	-	Nd–11	-	[46]
Caffeic acid	rice bran	4	3	89	[34]
	flour from polished rice	0.14	0.15	107	[34]
	brown rice flour	1.1 ^1^, 0.09 ^2^, 0.103 ^3^	0.7 ^1^, 0.05 ^2^, 0.26 ^3^	62 ^1^, 59 ^2^, 266 ^3^	[34] ^1^, [38] ^2^, [29] ^3^
	wheat flour	3.2	2.6	81	[29]
	oat flour	1.6	2.5	158	[29]
	buckwheat	Nd	Nd–0.03	47–99	[67]
	apple pomace, cv. Pink Lady	0.07	0.11–0.12	143–171	[68]
	corn flour/sorghum flour/apple pomace	5.0–5.7	4.2–4.9	84–86	[43]
p-Coumaric acid	corn grit	16–21 ^1^	18–21 ^1^, Nd ^2^	118–137 ^1^	[24] ^1^, [46] ^2^
	rice bran	4.0	4.2	103	[34]
	flour from polished rice	2.8	0.8	31	[34]
	brown rice flour	5 ^1^, 12 ^2^, 6 ^3^	4 ^1^, 14 ^2^, 6 ^3^	79 ^1^, 115 ^2^, 97 ^3^	[34] ^1^, [38] ^2^, [29] ^3^
	wheat flour	4	4	90	[29]
	oat flour	2.1	1.9	92	[29]
	buckwheat	0.03	0.04–0.10	122–322	[67]
	corn/kale, var. sabellica	-	0.13–0.17	-	[66]
	apple pomace, cv. Pink Lady	0.06	0.14–0.21	233–350	[68]
	corn flour/sorghum flour/apple pomace	1.0–1.5	0.5–0.9	46–63	[43]
	corn/red potato, var. Magenta Love	-	Nd–0.4	-	[46]
	rice/pea	-	0.03–0.08	-	[69]
Protocatechuic acid	corn flour/sorghum flour/apple pomace	2–5	0.2–1.6	10–29	[43]
	rice/pea	-	Nd–0.04	-	[69]
	buckwheat	0.4	0.2–0.3	47–75	[67]
Salicylic acid	buckwheat	0.25	0.04–0.11	18–46	[67]
	corn/kale, var. sabellica	-	0.02–0.04	-	[66]
	corn flour/sorghum flour/apple pomace	5–7	3–5	67–74	[43]
	rice/pea	-	0.03–0.08	-	[69]
3-OH-Cinnamic acid	corn/kale, var. sabellica	-	0.02	-	[66]
Cynarin and cynaroside	wheat/dried artichoke	1.8–5.4	0.6–1.3	25–34	[52]
	wheat flour	Nd	Nd	-	[52]
Diferulic acid	brown rice flour	7	8	118	[38]

The exponents refer to the literature in the last column; if the variety is not listed in the table, then either the data refer to more than one variety or the variety was not listed in the article; “-” means no measured data, Nd—below the detection limit.

**Table 4 foods-10-02100-t004:** Total flavonoid content (TFC) in the raw material and extrudates from different food materials.

Source	TFC in Input Raw Material (mg/100 g DM)	TFC in Output Extrudates (mg/100 g DM)	Retention (%)	Type of Extruder	Reference
rice bran	612	642	104	co-rotating twin-screw	[34]
polished rice	57	34	60	co-rotating twin-screw	[34]
brown rice	171	119	70	co-rotating twin-screw	[34]
corn	-	35	-	single-screw	[45]
corn/purple potato (var. Blue Star)	-	47–99	-	single-screw	[45]
corn/red potato (var. Magenta Love)	-	85–206	-	single-screw	[45]
lupin concentrate/lupin isolate/carrageenan	79	77	97	twin-screw	[7]
lupin concentrate/lupin isolate/carrageenan/semolina	79–142	77–183	120–129	twin-screw	[7]
millet	183	219	120	twin-screw	[56]

DM—dry matter; if the variety is not listed in the table, then either the data refer to more than one variety or the variety was not listed in the article; “-” means no measured data.

**Table 5 foods-10-02100-t005:** Total flavonols in the raw material and extrudates from different food materials.

Source	Total Flavonols in Input Raw Material (mg/100 g DM)	Total Flavonols in Output Extrudates (mg/100 g DM)	Retention (%)	Type of Extruder	Reference
apple pomace, cv. Pink Lady	21	26–28	122–133	co-rotating twin-screw	[68]
lentil flour	15	2	13	co-rotating twin-screw	[76]
corn	-	17	-	single-screw	[45]
corn + purple potato (var. Blue Star)	-	18–24	-	single-screw	[45]
corn + red potato (var. Magenta Love)	-	17–33	-	single-screw	[45]

DM—dry matter; if the variety is not listed in the table, then either the data refer to more than one variety or the variety was not listed in the article; “-” means no measured data.

**Table 6 foods-10-02100-t006:** Individual flavonol content in extrudates from different food materials prepared by twin-screw extruder.

Flavonol	Source	Flavonol Content in Input Raw Material (mg/100 g DM)	Flavonol Content in Output Extrudates (mg/100 g DM)	Retention (%)	Reference
Catechin hexoside	lentil flour with yeast	33–66	2–6	5–9	[76]
Kaempferol-O-desoxyhexoside-O-hexoside-O-rutinoside	lentil flour with yeast	1–3	0.4–0.6	13–34	[76]
Quercetin	apple pomace, cv. Pink Lady	0.04	0.5–1.3	1125–3200	[68]
Quercetin-3-O-glucoside	lentil flour with yeast	4–5	0.4–0.6	8–11	[76]
Quercetin-3-O-glucoside	apple pomace, cv. Pink Lady	4	5–6	133–147	[68]
Quercetin-O-hexoside	lentil flour with yeast	3–4	0.3–0.5	10–12	[76]
Quercetin-O-pentoside	lentil flour with yeast	5–6	0.5–0.8	8–12	[76]
Quercetin-O-pentoside	rice, var. Montsianell/beans, vars. NRVP 20064637 and # TOV 002319/carob fruit, vars. Negreta and Roja	Nd–22	Nd–3.4	0–31	[73]

DM—dry matter; if the variety is not listed in the table, then either the data refer to more than one variety or the variety was not listed in the article; “-” means no measured data, Nd—below the detection limit.

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
