# Peer review of "Changes in Phenolics during Cooking Extrusion: A Review"

_foods, 2021, doi:10.3390/foods10092100_

Round 1

Reviewer 1 Report

The paper includes the modifications that I suggested.

Author Response

Thank you for your comment.  Evzen Sarka

Reviewer 2 Report

The manuscript Changes of phenolics during cooking extrusion: A review fits the journal’s scope. The authors present the influence of extrusion on some phenolic compounds concentrations. The manuscript has been improved significantly, and the paper can be published after minor corrections (please see them below). Although not presented in two separate chapters, the flavonoid section has been systematized, and in the present form, is readable for other scientists.  

Minor

Tables 1, 2, 3 – Please add a note to show that in the original paper the authors didn’t indicate the variety

Tables 1, 2, 3 – please add a note below the table to explain “-” and Nd (table 3)

Author Response

Thank you for your comment, corrected.  Evzen Sarka

Reviewer 3 Report

The work takes into account previous review suggestions. 

Author Response

Thank you very much for your comment.  Evzen Sarka

This manuscript is a resubmission of an earlier submission. The following is a list of the peer review reports and author responses from that submission.

Round 1

Reviewer 1 Report

Despite the work that the paper reflects, major revisions are needed.

It is important to include the methodology of how the review has been conducted, as is done in other reviews of this same journal.

Line 26: In order not to interfere with the reading, it is more appropriate to put all the references at the end of the sentence

Line 27: The amount in grams per serving is included, but this information has no background. Complete it please.

Line 36: “destruction” is not adequate

Line 41: cite

Line 47: a temperature range is more suitable

Line 66-67: include more background

Line 71-end: some “co-workers” may be replaced by et al.

Line 135: Further develop possible alternatives to improve extrudates based on the materials being analyzed (rice, wheat ...)

In general, in section 2 it is necessary to explain why extrudates based on a single component are studied.

Section 3: The comment should be more organized and extensive, and include a brief explanation of the observed changes.

Section 4: It is necessary to increase the content and it should be related to the papers studied or the composition of the extruded products included in the Review

Tables:

Table 1: Reference A4 is not correct

Table 3: Rete-ntion, must be Retention

Some tables are unclear and difficult to understand

Each table looks different. Please, homogenize them.

Line 199: “… in literature[60]“ (for example) not “…in [60]”

References: the numbering of the references is duplicated and scientific names should be in italics.

Author Response

see in enclosure

Reviewer 2 Report

The manuscript Changes of phenolics during cooking extrusion: A review fits the journal’s scope. The authors present the influence of extrusion on some phenolic compounds concentrations. The critical approach of the present review is missing throughout the manuscript, and should be inserted. The extrusion process, and how it affects the stability of some phenolic compounds should be presented in more detail. TPC and TFC sections are well written and documented. However, a similar section to 3. Phenolic acids should be added for flavonoids.

The manuscript should answer if there are any other changes during the extrusion process (not only the lowering of concentration).

The discussion sections should be re-written (in this section are presented information that could/should be inserted in other sections – 2-4 and in the propose section for flavonoids).

Other comments

37-41 – the reference is missing

59-60 please correct

Table 3 – please present it on a single column (and not on two columns)

283-284, 314-315, 320-322 – please rephrase

311- please use italics for Latin names

Author Response

see in enclosure

Reviewer 3 Report

The manuscript "Changes of phenolics during cooking extrusion: A review" presents changes in the level of phenolic compounds in plant matrices under the influence of extrusion processes. The material was presented correctly, but the work requires minor corrections.

 Throughout the text, the entries should be corrected: mg / 100g DM should be mg / 100g-1 DM the improvement refers to lines 98, 103, 106, 111, 125, 130, 136, 252, 254, 257, 262, 266, 273 , 275, 288, 291 and 307 and entries in Tables 1, 2, 3, 4 and 5.

The text in lines 104-112 needs to be quoted

Notes on the content in table 1.

Add the parameters for extrusion of temperature, screw rotation, raw material moisture, and the method of determination (TPC)

Notes on the content in table 2.

Add parameters for extrusion of temperature, screw rotation, raw material humidity, and the method of determination (Total content of phenolic acids).

Notes on the content in Table 3.

Enter what method the phenolic acids were determined with

Notes on the content in table 4.

Add the methods of analysis (TFC) and extrusion parameters: temperature, screw rotation, raw material moisture.

Notes on the content in table 5.

Add total flavonols analysis methods and extrusion parameters: temperature, screw rotation, raw material moisture.

Author Response

see in enclosure

Round 2

Reviewer 1 Report

Despite the changes already made, I have to point out some suggestions to modify :

The comment on the results has been reorganized along with the discussion, for which paragraphs have been moved. I think that now it is more structured. Although the information has not been expanded in most cases as I indicated in the first review: “The comment should be more organized and extensive, and include a brief explanation of the observed changes.”

The numbering of the citations follows the old structure and not the current one. For example, line 131, reference 24 should be in order 23. It is necessary to correct the numbering of the citations in all the paper

Line 27: The amount in grams per serving is included, but this information has no background. Complete it please. – I am sorry, I don’t understand the notice. What are grams per serving?

The amount of breakfast cereals in grams that is a serving is fine, but it should have a little context, for example, in an average product, what amount of protein, fiber ... it contributes. If not, what is the ration value included for?

References: the numbering of the references is duplicated (where?.) and scientific names should be in italics. – sorry, what scientific names?, e.g. Moldavian dragonhead, Cocos nucifera, Brassica oleracea in Conclusions are in Italics

In the R1 version of the paper, the numbering of the final bibliography appeared in duplicate. In the new one it is corrected

The scientific names of the final bibliography are not in italics, for example reference 40. Oryza sativa

Author Response

see in enclosure

Reviewer 2 Report

The authors didn’t response to the majority of the comments. Thus, althought some modifications were made, the manuscript is hard to read and has many errors. The journal which the authors intend to publish is a scientific journal, and the information should be presented accordingly! (please see all your chapters!)

Row 62-63 – not all phenolics are bound to cell-wall components (please see other references as well).

My previous comment regarding the new section similar to section 3, was made not according to the number of lines, but on the following principles:

  1. TPC and TFC are non-specific determinations which are frequently used along with other assays (e.g. antioxidant activity);
  2. both, TPC and TFC are frequently used complementary to other methods (HPLC, HPTLC, etc.);
  3. mixing the results of TPC/TFC with other specific determinations (chromatography/spectrometry coupled methods) is not always correct;
  4. if the authors insist that section 3 and 4 represent a large part and section 5 was designed in order to be consistent (although is not a scientific argument), all three sections are only generally presented – a. the methods are not discussed (commonly, the TPC and TFC methods have several variants which can lead to different results – unlike chromatography in which the results are reproducible, at least in theory), b. the material used is biological, and therefore subject to great variability – thus, the cultivar/variety/hybrid should be mentioned for each type of material (rice, corn, wheat, etc – please see all your tables), but is not.

87-88 – please indicate the individual groups of phenolics

Author response: In addition, it can be found increased concentration and chemical changes, see all chapters.

Reviewer: Please indicate the lines for the parameters, except for the concentration

Line 354 – please use the scientific name of Moldavian dragonhead, red potatoes, strawberries, and apple pomace

Table 3 – please present it on a single column (and not on two columns)

Author Response

see in enclosure
